# Conservative Management and Ultrasound Follow-Up of Parasitic Myoma: Our Experience and Literature Review

**DOI:** 10.3390/diagnostics13101818

**Published:** 2023-05-22

**Authors:** Matteo Bruno, Erika Pelaccia, Christian Di Florio, Pierpaolo Palumbo, Laura Sollima, Manuela Ludovisi, Maurizio Guido

**Affiliations:** 1Obstetrics and Gynaecology Unit, San Salvatore Hospital, 67100 L’Aquila, Italy; 2Department of Life, Health and Environmental Sciences, University of L’Aquila, 67100 L’Aquila, Italy; 3Department of Diagnostic Imaging, Area of Cardiovascular and Interventional Imaging, Abruzzo Health Unit 1, 67100 L’Aquila, Italy; 4Unit of Pathology, San Salvatore Hospital, 67100 L’Aquila, Italy

**Keywords:** parasitic myoma, ultrasound, follow-up

## Abstract

A uterine fibroid is a benign smooth muscle neoplasm of the uterus. Parasitic fibroids (PMs) are a type of myoma that do not have any direct attachment to the uterus. PMs can arise from the implantation of tissue fragments generated during the morcellation process in previous laparoscopic myomectomies or hysterectomies. Transvaginal ultrasound (TV-US) may be helpful in the diagnosis of these benign tumors. Almost all the case reports in the literature suggest surgical management of parasitic fibroids. Conversely, we report an ultrasound-guided histological diagnosis and a non-surgical treatment of a parasitic myoma that arose twenty years after a total laparotomic hysterectomy and bilateral adnexectomy in a patient with multiple comorbidities, in order to offer a follow-up alternative to the management of this rare pathology. Our experience reveals that a non-surgical conservative approach can be considered as an alternative therapeutic option for the management of rare cases such as PMs in highly selected patients.

## 1. Introduction

A uterine fibroid or leiomyoma is a benign smooth muscle neoplasm of the uterus that characterizes women of reproductive age, with an estimated prevalence ranging from 4.5% to 68.6% [1]. However, its frequency is underestimated because in many women it is asymptomatic and remains undiagnosed [1]. When fibroids present symptomatically, the most common signs include abnormal uterine bleeding, pelvic pain/pressure, urinary disorders, constipation, and infertility. The International Federation of Gynecology and Obstetrics (FIGO) categorizes fibroids into eight subclasses, from type 1 to 8 [2]. Type 8 is a category that includes types such as the parasitic fibroids that do not have any direct attachment to the uterus and receive blood supply from another source [2]. A parasitic myoma (PM) was first described by Kelly and Cullen in 1909 as “myomathat has for some reason become partially or almost completely separated from the uterus” [3]; subsequently, Brody in 1953 described a similar entity attached to the omentum by a thin pedicle and a few blood vessels [4].

Usually, the location of parasitic fibroids is in the pelvis, close to the uterus; however, unusual locations such as the lungs, bladders, urethra, and sigmoid colon have also been reported [5].

PMs can be primary or iatrogenic [5]. Primary variants are presumed to derive from pedunculated subserous fibroid; they detach from the uterus and adhere to surrounding structures. The iatrogenic hypothesis postulates that parasitic fibroids arise from the implantation of tissue fragments generated during the morcellation process in previous laparoscopic myomectomies or hysterectomies [5]. According to the literature, recent developments in the field of minimally invasive surgery have led to an increase in the incidence of parasitic myomas; however, because the entity is still rare, the literature is limited to case reports or series [6]. 

Sometimes parasitic myomas are incidentally detected, being totally asymptomatic; as pedunculated fibroids, they can twist on the pedicle, causing lower abdominal discomfort, nausea [7], and pressure on the urinary or intestinal tract [8]. Abdominal pain is often sudden and severe, requiring immediate surgery [9,10]. From a clinical point of view, a parasitic myoma represents an interesting diagnostic problem, since it mimics other malignancies, and serious diagnostic errors may result as a consequence. The most useful modalities for detecting extrauterine leiomyomas are ultrasonography (US), computed tomography (CT), and magnetic resonance imaging [11]. Usually, due to accessibility in the outpatient setting, the initial imaging modality of choice is ultrasonography [12]. 

With US, a typical leiomyoma has variable echogenicity depending on the entity of degeneration (myxoid, hyaline, cystic), fibrosis, and calcification [11]. The differential diagnosis for parasitic leiomyomas includes masses of ovarian origin (both primary neoplasms and metastases), uterine malignant lesion-like sarcomas, broad ligament cysts, and lymphadenopathy; transvaginal ultrasound (TV-US) may be helpful, since it can show clear visual separation of the uterus and ovaries from the mass [11]. Ultrasonography can aid in the differential diagnosis between benign and malignant myometrial lesions [13]. In the 2019 multicenter study, Ludovisi et al. showed that an ultrasound detailing a large uterine myometrial tumor with inhomogeneous echogenicity, internal irregular cystic areas, increased vascularity, absence of shadows, and absence of calcifications suggested malignancy [13].

As highlighted in several case reports, management of parasitic fibroids is usually undertaken surgically, either via laparoscopy or open surgery [5,14,15]. 

Differently from standard management, we report on a ultrasound-guided tru-cut biopsy with histological diagnosis and a non-surgical treatment of a parasitic myoma that arose twenty years after a total laparotomic hysterectomy and bilateral adnexectomy in a patient with multiple comorbidities, in order to offer a follow-up alternative to the management of this rare pathology.

## 2. Case Report

A 66-year-old, multiparous woman was referred to our hospital for a follow-up US scan. This patient had normal stature and severe obesity, with a body mass index of 40 kg/m^2^.

The patient had been suffering from hypertension and type II diabetes mellitus, for which she was undergoing drug therapy. She also had a previous history of gastric cancer, which had been treated with gastroresection and Billroth-I reconstruction four years previous to the study. At 44 years old, she underwent a total laparotomic hysterectomy and bilateral adnexectomy for multiple uterine myomas. After the gynecological surgery, hormone replacement therapy (HRT) was administered to the patient for a period of 6 years.

On examination, her vitals were found to be stable. The abdomen was soft, nontender and not distended; the mass was not appreciated by palpation. The liver and the spleen were palpated as normal. A speculum examination of the vagina showed no abnormalities. The patient’s serum Ca 125 (9.2 U/mL) and HE4 (11.5 pmol/L) were within the reference range.

TV-US performed by an ultrasonoghrapher with advanced skills in the gynaecological field (M.L.), highlighted in the right pelvic cavity, near the pelvic wall, an intraperitoneal solid mass of 20 × 17 × 21 mm in size, with a spherical shape, echotexture with radial stripes, and shadowing (Figure 1, Appendix A). A Color Doppler examination showed no central and peripheral vascularization (Figure 2). No other lesions with the same characteristics were found. Due to benign ultrasound features, a single lesion and previous surgical history, the specific diagnosis via the ultrasound in the original ultrasound report according to pattern recognition suggested suspicion of a PM.

The patient underwent a CT scan of the abdomen that confirmed a 2-cm solid mass, iso-hypodense and with a regular outer contour, in the right pelvic cavity, medial to the hypogastric artery (Figure 3). There were no suspicions of peritoneal carcinomatosis or lymphnodal involvement.

Considering the risks associated with surgery and the suspected benign features of the lesion, an ultrasound tru-cut biopsy of the lesion was performed. The immunohistochemical features were compatible with hypocellular leiomyoma (Smooth Muscle Actin SMA+; Desmin+), characterized by a low mitotic count (1 mitotic figure per 20 high-power fields HPF) and a Ki-67 proliferative index of 1% (Figure 4). Immunohistochemical (IHC) staining of the myoma showed low expression of the estrogen receptor (ER) and no expression of the progesterone receptor (PR).

In agreement with the patient, and in consideration of the severe obesity, multiple previous surgeries, anesthesiological risk, and histology, we decided to perform conservative and non-surgical management.

At this point, the patient had been in follow-up with regular ultrasound checks, performed every three months, for one year. The volume of the lesion remained unchanged in subsequent checks.

## 3. Discussion

We report the ultrasound-guided histological diagnosis and a non-surgical treatment of a parasitic myoma that arose twenty years after a total laparotomic hysterectomy and bilateral adnexectomy in a patient with multiple comorbidities, in order to offer a follow-up alternative to the management of this rare pathology. Our case has some specific peculiarities: the disease occurred after a laparotomic hysterectomy, it was treated with a conservative approach, and it remained stable in the subsequent follow-up.

A surgical history of laparoscopic morcellation has been associated with PMs in the recent literature [14,15] due to the spread of the morcellated fragments inside the peritoneal cavity. However, cases of parasitic myoma after morcellation have been reported in the literature since 1997 [16]; indeed, there are several recent case reports concerning this phenomenon. In 2011, Cucinella et al. described four cases of parasitic myomas; in their study, all the patients had previously undergone laparoscopic myomectomy with the use of the morcellator to extract the lesion [17]. Recently, Lu et al. in their retrospective study of six patients with PMs reported that all patients had a history of laparoscopic hysterectomy or myomectomy with the use of a power morcellation [18].

A systematic review, published in 2015 by Van der Meulen et al., included five cohort studies that reported the overall incidence of parasitic myomas after laparoscopic surgery with the use of morcellation to be between 0.12 and 0.95% [6].

Conversely, cases reported in the literature where a PM develops after laparotomy or vaginal surgery are few. Unlike previous recent reports, the present case showed no history of laparoscopic surgery or intracavitary morcellation, and it also showed the necessity of consider other factors; indeed, the finding of a PM is not a frequent occurrence, whereas the loss of tissue fragments during morcellation is presumably extremely common [17].

For this reason, it can be assumed that there are other factors that increase the risk of this complication [17]; exposure of fragments to steroid hormones and growth factors may play a role [19]. This idea is also suggested in the case treated by Takeda et al. in 2011 [20], where a 1.4 cm in diameter mass in the retrovesical area occurred in a 27-year-old patient with a history of laparoscopic-assisted myomectomy and previous extraction of the myoma node performed via tissue morcellation with a cold knife. The myoma, which had not changed in size for two years, grew rapidly during early pregnancy, suggesting the important role of hormones in this growth [20].

In our specific case, the patient had taken hormone replacement therapy (HRT) for a long period: this probably suggest the role of hormones in the risk of development or growth of the PM.

In almost all the cases reported in the literature, the management of parasitic myomas is always a surgical approach. All parasitic myomas diagnosed in the 69 women identified in Van der Meulen’s systematic review were removed surgically, despite 21.7% of women being asymptomatic [6]. Moreover, our conservative approach with ultrasound diagnosis and subsequent follow-up differs from common reports and has no antecedents: this choice emerges from the patient-related comorbidities and the complete absence of reported symptoms. Nonsurgical management and ultrasound follow-up can be applied to highly selected cases: although a PM is usually a benign mass, cases of malignant transformation have been reported [21]; the risk of malignant transformation with PMs is higher than that with uterine myomas (2–5% versus 2–3% hysterectomies for uterine myomas) [22,23]. Despite the limitation due to false negatives in the histological analysis of the tissue, an ultrasound tru-cut biopsy of the lesion allows us to have a diagnosis, as well as to modify the therapeutic approach if the ultrasound or clinical conditions should change.

In our patient, the instrumental exams and histopathological characteristics were highly suggestive of a pelvic benign mass, indicated by the absence of cellular atypia, tumor cell necrosis, or increased mitotic figures, in addition to the benign ultrasound features and the high anesthesiologist risk due to the patient’s comorbidities; all these characteristics guided us in the therapeutic choice. Indeed, some rare benign types of leiomyomas, such as cellular leiomyomas or mitotically active leiomyomas, could be confused with a sarcoma [13].

An important role was played by the ultrasound examination performed by an ultrasonoghrapher with advanced skills in the gynecological field, which allowed us to make an easy diagnosis of a PM, even one of a small dimension. Di Legge et al. demonstrated that, even if the lesion is very small, its ultrasound features can still help in the diagnosis [24].

However, in some cases, clinicians and surgeons are not able to make a preoperative diagnosis of a PM, but the investigation could still be highly suggestive of malignancy (large heterogeneous masses, raised tumor markers, and lymphadenopathy), so the surgical treatment is justified [25].

The present case report is the result of our personal experience; because of the lack of studies on conservative management in PMs, our report represents an isolated case and emerges from the choice of personalized treatment. Despite that, it is our opinion that our research can be a useful source for a deeper investigation of other expert center groups.

## 4. Conclusions

Our study considers a non-surgical conservative approach as an alternative therapeutic option for the management of rare cases, such as PMs in highly selected patients. The personalization of the strategy and the achievement of a diagnostic histology for PMs should be the aim of treatment. Despite the limits due to false-negative biopsies, an ultrasound-guided tru-cut biopsy could play a role in establishing a diagnosis that allows appropriate referral of the patient to a specialized center that provides multidisciplinary care of these rare tumors and where a conservative approach instead of surgery may be adopted. Ultrasound is also an optimal modality for the surveillance of benign PMs.

To date, knowledge on the behavior of PMs is very limited, therefore further studies will allow for more adequate knowledge in the management of patients whose general incidence appears to be increasing.

## Figures and Tables

**Figure 1 diagnostics-13-01818-f001:**
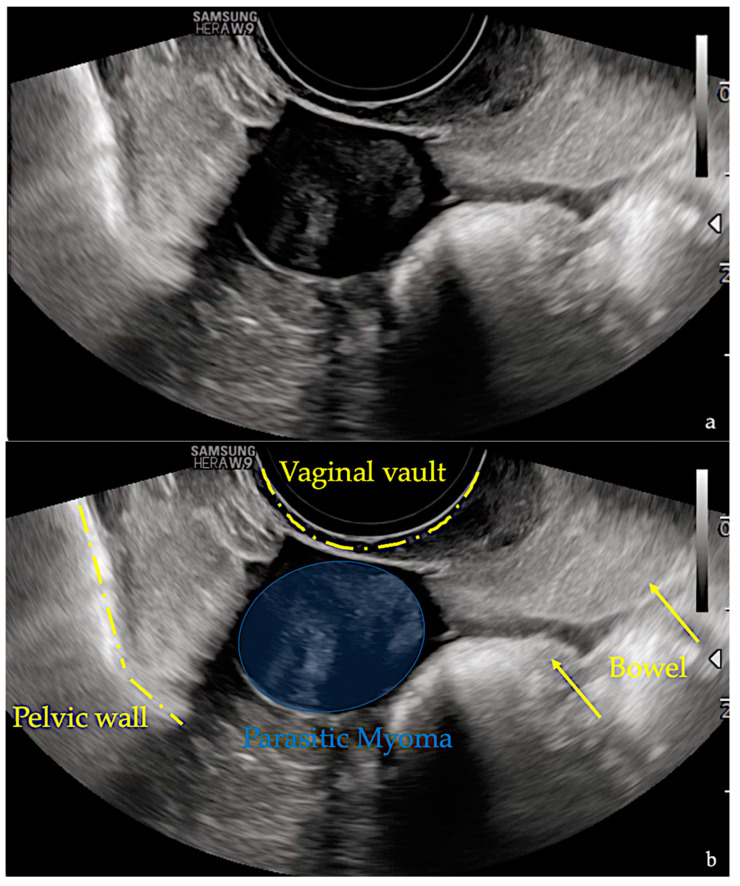
Images (**a**,**b**) show ultrasound images performed at the time of diagnosis. The images demonstrate a well-defined spherical intraperitoneal mass of 20 × 17 × 21 mm in size. Echotexture with radial stripes and shadowing are well represented. No other similar lesions were detected. In image b, the pelvic structures are highlighted.

**Figure 2 diagnostics-13-01818-f002:**
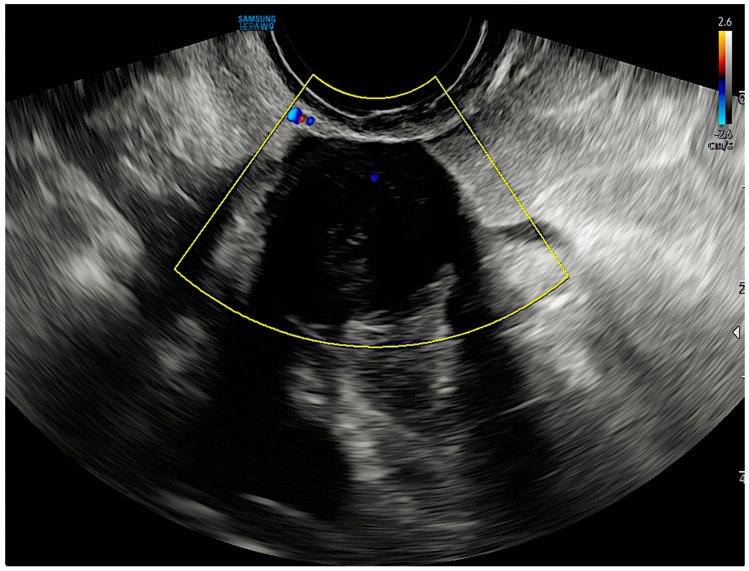
Transvaginal ultrasound image shows no central and peripheral vascularization during the Color Doppler examination.

**Figure 3 diagnostics-13-01818-f003:**
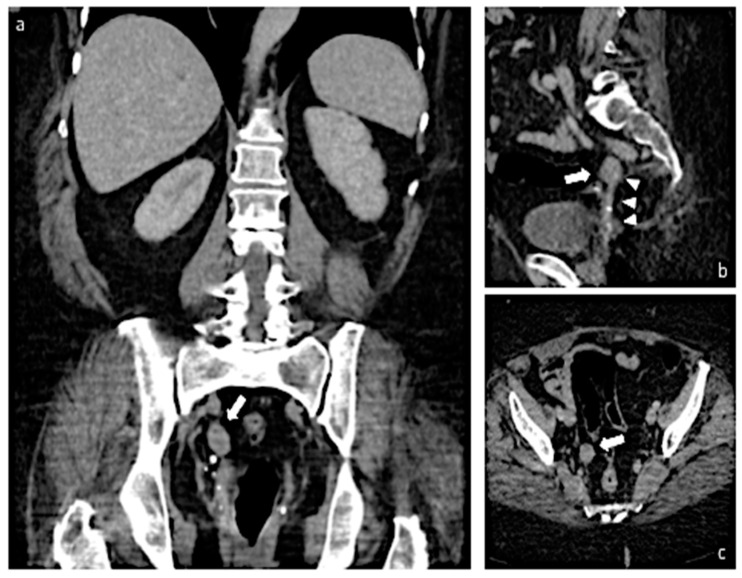
Images (**a**–**c**) show MPR images of a portal venous CT acquisition. In all images, thick white arrows highlight a hypovascular nodule located in the right hemiportion of the pelvic cavity (image (**a**), coronal view; image (**b**), sagittal view; image (**c**), axial view). In image (**b**), white arrowheads point out a pedicle connecting with the vaginal stump. MPR: multiplanar reconstruction.

**Figure 4 diagnostics-13-01818-f004:**
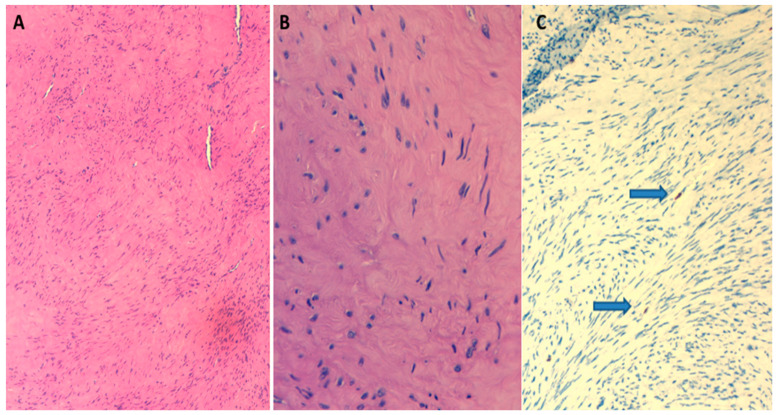
Hematoxylin and Eosin, 10×. Tumor composed of spindle cells arranged in intersecting fascicles (image (**A**)). Hematoxylin and Eosin, 40×. The spindle cells have bland cytological features with elongated nuclei and eosinophilic fibrillary cytoplasm (image (**B**)). Ki67/MIB1 immunostaining 20×. Low proliferation index (arrows: positive nuclei) (image (**C**)).

## Data Availability

Data sharing not applicable. No new data were created or analyzed in this study. Data sharing is not applicable to this article.

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
