# Peer review of "Conservative Management and Ultrasound Follow-Up of Parasitic Myoma: Our Experience and Literature Review"

_diagnostics, 2023, doi:10.3390/diagnostics13101818_

Round 1

Reviewer 1 Report

Esteemed Authors and Editor,

I think the theme of the article is quite interesting and the manuscript well written. I fully support its being published.

I have some minor comments:

- at lines 67-70, where US suspicion of malignancy is explained, there is no mention of increased vascularity as being one of the criteria (for either STUMP or malignancy).

- perhaps the technical difficulties of the biopsy should be addressed, considering the myoma had no attachments and it was free floating in the pelvis? At the same time, perhaps confirmation as a myoma by MRI (and not CT, which does not prevail for reproductive pelvic pathology) could suffice. In no other scenario do we feel the need to perform biopsy to confirm myoma. 

- some amount of free peritoneal fluid is seen in Figure 1, not too much, but more than the usual. Was this constant at follow-up examinations?

- was the histology from hysterectomy available to check if the PM had the same structure as the initial fibroids?

- I wonder how fragments of myoma could end up in the peritoneal cavity at abdominal hysterectomy. Perhaps if prior to removing the uterus the fibroid had been enucleated to help dissection. This should be checked with the hysterectomy protocole, if available and deserves some comments.

- the surgical approach is mentioned as recommended for PMs. However, I think this should rather be tailored according to fibroid size, and patient age an comorbidities, such as in the present case.

- no-surgical should be changed to non-surgical along the manuscript.

Reviewer 2 Report

This is a case report, where authors  report the ultrasound-guided histological diagnosis and a no-surgical treatment of a parasitic myoma that arise twenty years after a total laparotomic hysterectomy and bilateral adnexectomy in a patient with multiple comorbidities. They conclude that no-surgical conservative approach can be considered an effective therapeutic option for the management of rare cases such as PMs in highly selected patients.

This is a well written case report, although the paper needs corrections in language and grammar. in the end of the introduction, the rationale should be more persuasive. 

Data accompanied are welcome.

The discussion section needs reformatting, e.g. the first paragraph should contain the findings, the next ones the comparisons with similar papers.

The need for an accurate evaluation -for fear of malignancy- should also be highlighted, together with the false negative results after a biopsy. Finally, the need for close surveillance after conservative management, especially, in this case where other comorbidities exist.

The phrase «our experience suggests:” should be changed, as the report is for only one case. I also believe that authors should be more careful in their proposal that parasitic myomas could be treated conservatively, and more conservative with their conclusion.

please check my comments above
